# Possibility of deterioration of respiratory status when steroids precede antiviral drugs in patients with COVID-19 pneumonia: A retrospective study

Yu Shionoya[1], Toshibumi Taniguchi[2], Hajime Kasai[1,3]*, Noriko Sakuma[1], Shun Imai[1], Kohei Shikano[1], Shin Takayanagi[4], Misuzu Yahaba[2], Taka-aki Nakada[5], Hidetoshi Igari[2], Seiichiro Sakao[1], Takuji Suzuki[1]

1 Department of Respirology, Graduate School of Medicine, Chiba University, Chiba, Japan, 2 Department of Infectious Diseases, Chiba University Hospital, Chiba, Japan, 3 Health Professional Development Center, Chiba University Hospital, Chiba, Japan, 4 Matsudo City General Hospital Department of Infectious Diseases, Matsudo, Japan, 5 Department of Emergency and Critical Care Medicine, Chiba University Graduate School of Medicine, Chiba, Japan

* daikasai6075@yahoo.co.jp

**Data Availability Statement:** All relevant data are within the manuscript and its Supporting Information files.

## Abstract

### Introduction

Coronavirus disease (COVID-19) is caused by severe acute respiratory syndrome coronavirus 2. Although most patients with COVID-19 develop asymptomatic or mild disease, some patients develop severe disease. The effectiveness of various therapeutic agents, including antiviral drugs, steroids, and anti-inflammatories for COVID-19, have been being confirmed. The effect of administering steroids in early disease is unclear. This study therefore aimed to evaluate the effectiveness and risk of exacerbation of steroids administered preceding antiviral drugs in patients with COVID-19 pneumonia.

### Methods

This retrospective, single-center, observational study included consecutive patients with COVID-19 between March 2020 and March 2021. Patients were divided into a steroids-first group and antiviral-drugs-first group. Mortality, duration of hospitalization, incidence rate and duration of intensive care unit (ICU) admission, intubation, and extracorporeal membrane oxygenation (ECMO) induction of the two groups were compared.

### Results

A total of 258 patients were admitted during the study period. After excluding patients who received symptomatic treatment only, who were taking immunosuppressive drugs, or who were administered antiviral drugs only, 68 patients were included in the analysis, 16 in the steroids-first group and 52 in the antiviral-drugs-first group. The rate of intubation, ICU admission and ECMO induction were significantly higher in the steroids-first group than in the antiviral-drugs-first group (81.3% vs. 33.3, *p*<0.001, 75.0% vs. 29.4%, *p* = 0.001, and

**Funding:** The authors received no specific funding for this work.

**Competing interests:** T.T. has received honorarium for lecture from Gilead Sciences Inc., Janssen Pharmaceuticals Inc., ViiV Healthcare Limited, and MSD Limited, respectively. T.T. is the member of advisory board of Janssen Pharmaceuticals Inc., ViiV Healthcare Limited, and MSD Limited, respectively. This does not alter our adherence to PLOS ONE policies on sharing data and materials.

31.3% vs. 7.8%, $p = 0.017$, respectively). Furthermore, patients who received steroids within ten days after starting antiviral drugs had significantly lower rates of ICU admission, intubation, and ECMO induction. (81.3% vs. 42.9% $p = 0.011$, 75.0% vs. 37.1% $p = 0.012$, and 31.3% vs. 8.6% $p = 0.039$, respectively).

## Conclusions

Administering steroids prior to antiviral drugs soon after symptom onset can aggravate disease severity. When administration of steroids is considered soon after symptom onset, it may be safer to initiate antiviral drugs first.

## Introduction

Coronavirus disease (COVID-19), caused by severe acute respiratory syndrome coronavirus 2 (SARS-CoV-2) spread rapidly worldwide from February 2020, and on March 11, 2020, the World Health Organization declared it a pandemic [1]. Although the majority of patients with SARS-CoV-2 infection are asymptomatic or develop mild disease, 14% develop severe disease and 5% develop critical disease [2].

The effects of various therapeutic agents for COVID-19, including antiviral drugs, steroids, and anti-inflammatories had been verified by March 2021. Dexamethasone and tocilizumab: anti-interleukin-6 receptor monoclonal antibody, have been shown to be clinically effective for COVID-19 [3,4]. The effect of dexamethasone was reported in the Randomized Evaluation of COVID-19 Therapy (RECOVERY) open-label trial [3]. In patients hospitalized with COVID-19, the use of dexamethasone resulted in a lower 28-day mortality among patients who received either invasive mechanical ventilation or non-invasive supplemental oxygen. However, the results of the RECOVERY trial did not mention any apparent significance of dexamethasone given within 7 days of onset in COVID-19. Thus, the timing of drug administration, including antiviral drugs, is unclear and the most effective therapeutic approach has not been determined. Additionally, in Japan, the results of the RECOVERY trial led to the insurance approval of dexamethasone, although the Japanese guidelines for COVID-19 treatment do not provide definite rules regarding the timing of administration of dexamethasone. As a result, dexamethasone tends to be liberally administered at the discretion of clinicians, even in early onset or mild cases of COVID-19.

COVID-19 exhibits three phases of increasing severity [5,6]. During the first phase (viral response phase) SARS-CoV-2 enters susceptible host cells by binding to human angiotensin-converting enzyme 2 receptors. In this phase, patients are asymptomatic or develop mild symptoms, such as fever, cough, and loss of smell and/or taste. About 20% of individuals develop severe disease with dyspnea (defined as a $PaO_2/F_IO_2$ <300 mmHg). This second phase (viral response and host inflammatory response overlap phase) is characterized by pulmonary disease, viral multiplication, and localized inflammation in the lungs. The third phase (host inflammatory response phase), which is characterized by cytokine storms, may occur 7–8 days after symptom onset. In the RECOVERY Trial, dexamethasone was found to be beneficial to patients who were treated more than 7 days after symptom onset [3]. The timing of the initiation of steroids might be important in COVID-19. Cantini et al. [6] reported that a meta-analysis showed that combination therapy was more effective than monotherapy for treating COVID-19. For example, the administration of antiviral drugs in the viral response

phase and steroids and anti-inflammatory drugs in the host inflammatory response phase was found to be an effective combination.

As of March 2021, in Japan, the mainstay of treatment for COVID-19-associated pneumonia was remdesivir, favipiravir, dexamethasone, and tocilizumab. Among these drugs, remdesivir and dexamethasone are covered by insurance, while favipiravir and tocilizumab are not [7]. Although remdesivir has been covered by insurance since May 7, 2020, application to the Ministry of Health, Labor and Welfare for remdesivir use is required because of limited supply. On the other hand, dexamethasone, which has long been used to treat multiple diseases, is inexpensive and readily available. Therefore, dexamethasone has been used extensively for COVID-19 treatment in Japan.

This study aimed to evaluate the effectiveness and risk of exacerbation of COVID-19 associated pneumonia when steroid initiation preceded antiviral drug initiation.

## Material and methods

### Study design and materials

This retrospective, single-center, observational study included consecutive patients with COVID-19 admitted to our institution between March 2020 and March 2021. Patients who received only symptomatic treatment or who were originally treated with steroids or immunosuppressive drugs because of complications were excluded.

The study was approved by the ethics committee of Chiba University, Japan (approval number 3929). The requirement for informed consent was waived because of the study had a retrospective design. The source of the data analyzed in this study was the medical records, Chiba University Hospital.

All patients were suspected to have COVID-19 based on symptoms such as fever, cough, and dyspnea and/or a history of contact with a person with confirmed COVID-19 were tested for SARS-CoV-2. The diagnosis was confirmed by SARS-CoV-2 testing using a polymerase chain reaction assay.

In our institution, we evaluated the disease severity based on fever, severity of respiratory failure based on oxygen saturation, and the presence of pneumonia. Known risk factors for exacerbations include age, uncontrolled diabetes, cardiovascular diseases, and obesity [8]. According to severity and risk factors, antiviral drugs (favipiravir or remdesivir) were initiated first. Then, if respiratory failure worsened, steroids (mostly dexamethasone) were administered, and if patients needed a fraction of inspiratory oxygen > 0.4 at the time of admission, antiviral drugs and dexamethasone were initiated at almost the same time, but antiviral drugs were initiated 3 to 6 hours ahead of dexamethasone. Furthermore, tocilizumab was administered in patients who developed severe respiratory failure that did not respond to antiviral drugs and steroids. According to our institute's protocol, when the D-dimer level was > 3.0 mg/dL, we administered direct oral anticoagulants in non-ICU patients, but withheld anticoagulants in elderly patients and patients with a hemorrhagic diathesis (two patients). We continued aspirin in in non-ICU patients who had originally taken aspirin (two patients). There were no patients who had originally taken direct oral anticoagulants. We administered heparin routinely to ICU patients. Some patients transferred to our institution from other hospitals for intensive or specialized treatment had received steroids prior to being transferred.

Patients were divided into two groups: a steroids-first group and an antiviral-drugs-first group (administered antiviral drugs first and then administered steroids). In each group, the mortality, duration of hospitalization, rate, and duration of intensive care unit (ICU) admission, intubation, and extracorporeal membrane oxygenation (ECMO) induction were

compared. Data were also collected on comorbidities and the results of blood tests performed on the day of admission.

## Statistical analysis

The results of continuous variables were expressed as the mean ± standard deviation. The continuous variables were compared between the groups using the Mann-Whitney $U$ test and the proportions of categorical variables were compared using the chi-square test. The Kaplan-Meier method was used to determine the ICU admission rate, intubation rate, ECMO introduction rate and 30-day mortality rate. The ICU admission rate, intubation rate, ECMO introduction rate and survival estimates were compared using the log-rank test. The threshold for statistical significance was set at $p = 0.05$. All statistical analyses were performed using JMP Pro 14 software (SAS Institute Inc., Cary, NC, USA).

# Results

A total of 258 patients were admitted to the hospital with COVID-19 during the study period. Of these patients, 129 were excluded because they received only symptomatic treatment (n = 117) or were originally using steroids or immunosuppressive drugs because of complications (n = 12). The129 remaining patients were administered specific treatment for COVID-19. Sixty-one patients who were treated with only antiviral drugs such as favipiravir and/or remdesivir were excluded. Furthermore, one patient who administered antiviral drugs and tocilizumab without steroids was excluded. Finally, sixty-seven patients were included in the analysis (Fig 1). Of the 67 patients included in the analysis, 16 and 51 were in the steroids-first and the antiviral-drugs-first groups, respectively.

## Patient characteristics

Table 1 summarizes the patient characteristics according to group. The antiviral-drugs-first group was significantly older than the steroids-first group (64.2±13.0 years vs. 57.8±13.2 years, $p = 0.050$), but there were no significant differences in the sex, body mass index, obesity, and prevalence of comorbidities between groups. The white blood cell count was significantly lower in the antiviral-drugs-first group than in the steroids-first group (8576.5 ± 6410.4 cells/µL vs. 11043.8± 3,704.2 cells/µL, $p = 0.001$). Furthermore, the lymphocyte percentage of white blood cells was significantly greater in the antiviral-drugs-first group than in the steroids-first group (13.5 ± 10.0% vs. 2.8 ± 1.7%, $p < 0.001$).

## Comparison of the outcomes among the steroids-first group and antiviral-drugs-first groups

Table 2 summarizes the treatment and course of disease according to group. Fig 2 shows the patient data on the rate of ICU admission, intubation, ECMO induction, and survival in the two study groups. The interval from symptom onset to steroid initiation was significantly shorter in the steroids-first group than the antiviral-drugs-first group (5.6±2.4 days vs. 9.7±5.6 days, $p<0.001$). The rate of intubation, ICU admission and ECMO induction were significantly higher in the steroids-first group than the antiviral-drugs-first group (81.3% vs. 33.3, $p<0.001$, 75.0% vs. 29.4%, $p = 0.001$, and 31.3% vs. 7.8%, $p = 0.017$, respectively). In contrast, there was no significant difference in the duration of ICU admission, or the duration of intubation and ECMO between the two groups.

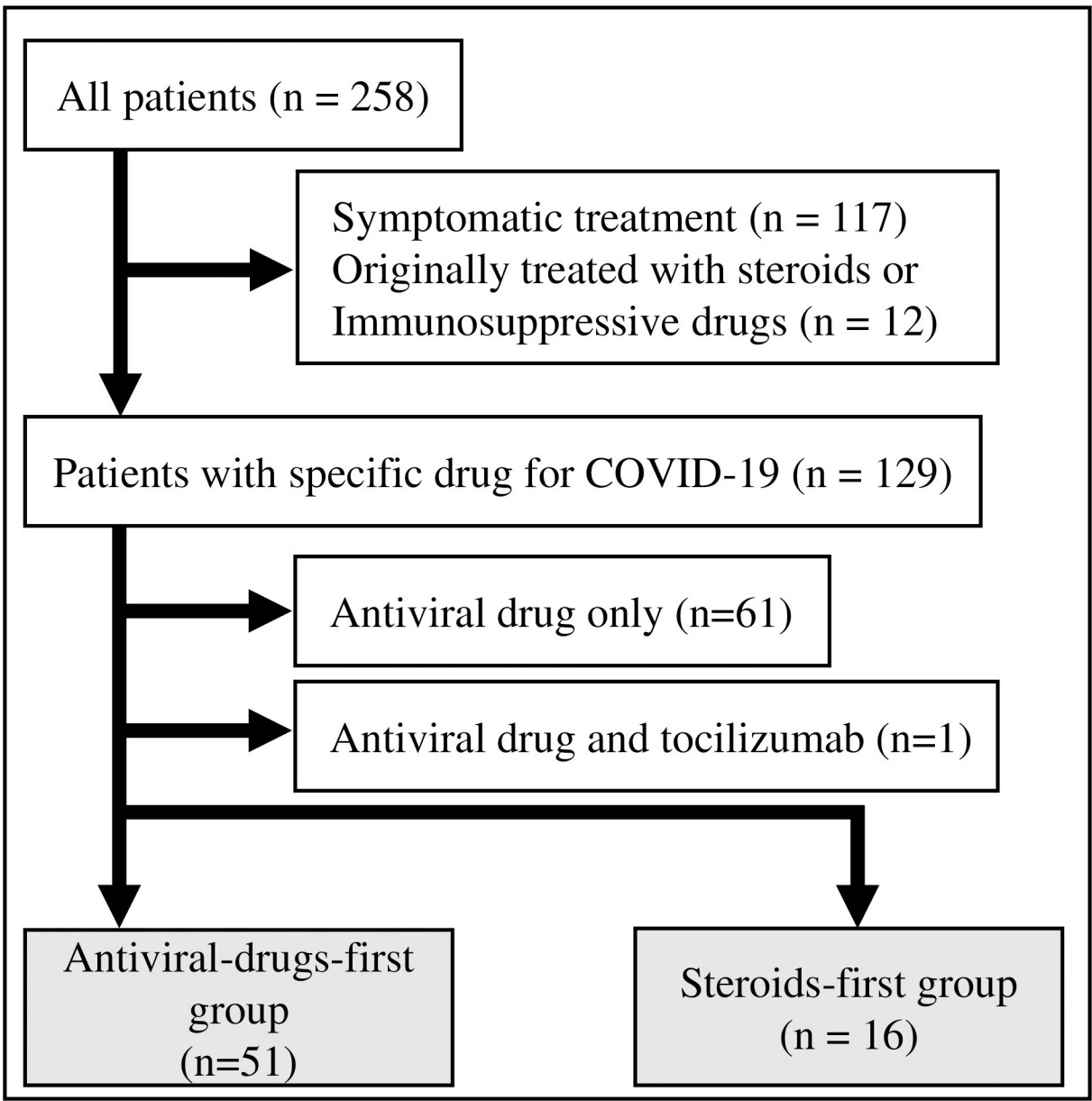

**Fig 1. Patient flow diagram showing the reasons for exclusion from the analysis.**

## Discussion

This study evaluated the effect of steroid administration in the early phase of the disease, prior to antiviral drug initiation, among patients hospitalized with COVID-19. The results suggest that early administration of steroids prior to antiviral drug administration may lead to further deterioration of respiratory status, which may increase the rate of ICU admission, intubation, and ECMO induction. Conversely, initiation of steroids after antiviral drugs may prevent respiratory deterioration even in the early days after symptom onset.

Older age is known to be a risk factor exacerbating COVID-19 [8]. The antiviral-drugs-first group were significantly older than the steroids-first group. Despite the antiviral drugs-first group being older than the steroid-first group, the antiviral drugs-first group had a better

**Table 1. Baseline characteristics (N = 67).**

| Parameter | All (N = 67) | antiviral-drugs-first group (N = 51) | steroids-first group (N = 16) | P-value |
|---|---|---|---|---|
| Age, years | 62.6 ± 13.3 | 64.2±13.0 | 57.8±13.2 | **0.050** |
| Sex (Male / Female) | 54/13 | 42/9 | 12/4 | 0.516 |
| Body mass index, kg/m$^2$ | 26.7 ± 6.9 | 26.4 ± 7.3 | 27.6 ± 5.8 | 0.285 |
| Comorbidity | | | | |
| Diabetes, n (%) | 36 (53.7) | 29 (56.9) | 7 (43.8) | 0.359 |
| Cardiovascular diseases, n (%) | 14 (20.9) | 11 (21.6) | 3 (18.8) | 0.809 |
| Hypertension, n (%) | 36 (53.7) | 29 (56.9) | 7 (43.8) | 0.359 |
| Dyslipidemia, n (%) | 16 (23.9) | 10 (19.6) | 6 (37.5) | 0.143 |
| Cancer, n (%) | 1 (1.5) | 1 (2.0) | 0 (0.0) | 0.573 |
| Blood examination | | | | |
| Date of testing from date of onset | 8.3 ± 3.7 | 7.9 ± 3.9 | 9.5 ± 2.8 | 0.092 |
| White blood cell, /μL | 9,165.7 ± 5,937.6 | 8,576.5 ± 6,410.4 | 11,043.8 ± 3,704.2 | **0.001** |
| Lymphocyte, % | 11.0 ± 9.9 | 13.5 ± 10.0 | 2.8 ± 1.7 | **<0.001** |
| C-reactive protein, mg/dL | 10.2 ± 8.1 | 9.8 ± 7.7 | 12.0 ± 9.4 | 0.394 |
| Lactate dehydrogenase, IU/L | 435.0 ± 179.0 | 432.8 ± 192.3 | 457.7 ± 123.6 | 0.314 |
| Ferritin, ng/mL | 795.8 ± 632.3 | 714.9 ± 535.9 | 1104.6 ± 876.9 | 0.160 |
| D-dimer, μg/mL | 5.8 ± 14.6 | 4.4 ± 10.4 | 11.6 ± 25.5 | 0.092 |

Data are presented as mean ± standard deviation.

**Table 2. Overview of treatments for COVID-19 and course of the disease (n = 67).**

| Parameter | All (N = 67) | antiviral-drugs-first group (N = 51) | steroids-first group (N = 16) | P-value |
|---|---|---|---|---|
| Duration of hospitalization, days | 25.1 ± 22.5 | 25.4 ± 24.4 | 23.9 ± 15.5 | 0.746 |
| Date of onset at the time of hospitalization, days | 8.3±3.7 | 7.8 ± 3.9 | 9.5 ± 2.8 | 0.076 |
| Therapeutic drugs | | | | |
| Favipiravir, n (%) | 38 (56.7) | 32 (62.8) | 6 (37.5) | 0.075 |
| The administration day from onset | 6.7 ± 2.2 | 6.8 ± 2.3 | 6.0 ± 1.1 | 0.283 |
| Remdesivir, n (%) | 45 (67.1) | 32 (62.8) | 13 (81.3) | 0.169 |
| The administration day from onset | 10.2 ± 5.1 | 10.1 ± 5.5 | 10.4 ± 3.9 | 0.528 |
| Steroids, n (%) | 67 (100.0) | 51 (100.0) | 16 (100.0) | - |
| The administration day from onset | 8.7 ± 5.3 | 9.7 ± 5.6 | 5.6 ± 2.4 | **<0.001** |
| Tocilizumab, n (%) | 25 (37.3) | 19 (37.3) | 6 (37.5) | 0.986 |
| The administration day from onset | 9.7 ± 2.7 | 9.8 ± 2.8 | 9.2 ± 2.5 | 0.724 |
| Anticoagulants or aspirin | 45 (67.2) | 32 (62.8) | 13 (81.3) | 0.169 |
| ICU admission, n (%) | 30 (44.8) | 17 (33.3) | 13 (81.3) | **<0.001** |
| Duration of ICU, days | 20.6 ± 20.3 | 24.0 ± 23.6 | 16.2 ± 14.6 | 0.249 |
| Intubation, n (%) | 27 (40.3) | 15 (29.4) | 12 (75.0) | **0.001** |
| Duration of intubation, days | 20.5 ± 20.7 | 23.7 ± 24.3 | 16.5 ± 15.1 | 0.298 |
| ECMO, n (%) | 9 (13.2) | 4 (7.8) | 5 (31.3) | **0.017** |
| Duration of ECMO, days | 14.6 ± 10.4 | 14.5 ± 12.4 | 14.6 ± 10.0 | 0.618 |
| Mortality, n (%) | 10 (14.9) | 7 (13.7) | 3 (18.8) | 0.623 |

ECMO, Extracorporeal membrane oxygenation. ICU, Intensive care unit.

Data are presented as mean ± standard deviation.

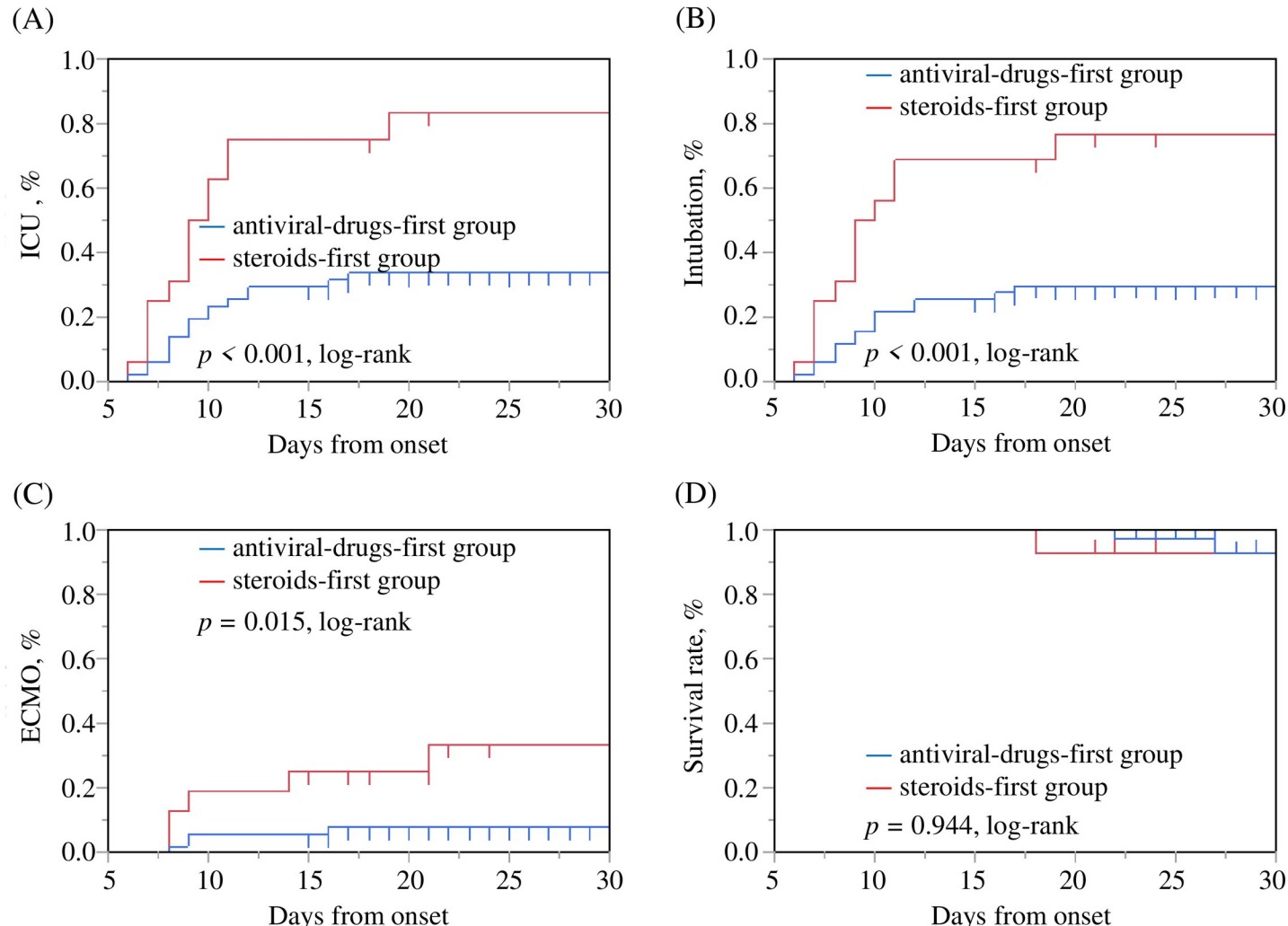

**Fig 2. Rate of ICU admission, intubation, ECMO induction, and mortality according to group.** The rate of ICU admission, intubation, ECMO induction were poorer in patients of the steroids-first group than in those the antiviral-drugs-first group ($p<0.001$, $p<0.001$ and $p = 0.015$, respectively, log-rank test). ECMO, Extracorporeal membrane oxygenation. ICU, Intensive care unit.

prognosis. The steroid-first group patients were administered steroids before admission to our hospital, therefore their lymphocyte count was significantly lower than that of antiviral drugs-first group. The blood tests performed on admission to our hospital were analyzed, because of the blood test results performed in the original hospital were frequently incomplete.

In this study, remdesivir (n = 45, 67.1%) and favipiravir (n = 38, 56.7%) were used and other antiviral drugs, such as lopinavir/ritonavir, were not. Several antiviral drugs have been assessed for COVID-19 [9], but remdesivir is the only Food and Drug Administration-approved drug for the treatment of COVID-19, based on the result of ACTT-1 trial [10]. Routine use of remdesivir is not recommended according to the results of some studies and the National Institutes of Health guidelines [11–13]. In cases where patients are at a particularly high risk of clinical deterioration or require minimal supplemental oxygen, remdesivir administration should be considered [11]. Our institute's protocol is to initiate antiviral drugs according to the severity and risk factors. In this study, the differences in the antiviral drugs

used may have affected the prognosis; however, there is a lack of strong evidence regarding the benefits of antiviral drugs in COVID-19.

In patients with COVID-19, steroid administration in the early phase of the disease, prior to initiation of antiviral drugs may be associated with a worsening of respiratory status. In this study, the steroids-first group had a significantly higher rate of intubation, ICU admission, and ECMO induction. In the RECOVERY Trial, only three of 2104 patients (0.14%) were administered antiviral drugs before being randomized [3]. Therefore, in almost all cases, patients were treated with dexamethasone prior to antiviral drug initiation. A subgroup analysis was performed comparing participants in the RECOVERY Trial in whom dexamethasone was initiated less than 7 days after symptom onset with participants in whom dexamethasone was initiated more than 7 days after symptom onset. The efficacy of dexamethasone administration less than 7 days after disease onset was not reported. As stated previously, COVID-19 has a viral response phase and a host inflammatory response phase [5,6]. Cytokine storms may occur 7–8 days after symptom onset, during the host inflammatory response phase. In this study, patients in steroids-first group were administered steroids relatively early in the course of the disease, within ten days after symptom onset. Some studies of SARS or Middle East respiratory syndrome found that steroid use may delay virus clearance and be associated higher viral concentrations [14,15]. Furthermore, Lee et al. [15] did not recommend early steroid use without effective antiviral drugs in SARS. However, the study by Lee et al. was not about COVID-19 and some studies reported that the administration of steroids to patients with COVID-19 may not affect clearance of SARS-CoV-2 [16–21]. Evidence of the effect of steroid use on clearance of SARS-CoV-2 is currently inconclusive. Some reports showed the viral load may correlate with the severity of COVID-19 [22,23]. A higher viral load leads to apoptosis of pneumocytes and endothelial cells, which in turn activates platelets and induces coagulation factors [24]. COVID-19 has a high rate of thromboembolic complications [25], and thromboembolic complications are associated with poor prognosis [26]. Furthermore, Mishra et al. [27] pointed out that steroid use itself may be associated with thromboembolic complications because steroids tend to increase clotting factors and fibrinogen concentrations. This pathophysiology in COVID-19 is poorly understood. Early administration of steroids may be associated with a higher viral load, leading to an exacerbation of the respiratory condition, and a possible increase in the risk of thromboembolic complications, and so steroid use may increase the need for critical care in some patients with COVID-19.

In a comparison of patients in whom steroids were initiated after antiviral drug initiation less than ≤10 days or ≥11 days after symptom onset, the former group had a significantly higher rate of ICU admission and tended to have a higher rate of intubation (S1 Table and S1 Fig). Thus, some patients should be administered steroid in the early days since symptoms onset because of worsening of respiratory status rapidly. While patients who were administered steroids less than ten days after symptom onset in antiviral-drugs-first group had a significantly lower rate of ICU admission, intubation, and ECMO induction than patients in the steroids-first group (S2 Table and S2 Fig). Furthermore, patients who received short-term steroids also had a significantly lower rate of ICU admission, intubation, and ECMO induction than patients in the steroid-first group (S3 Table). Thus, the order of drug administration may lead to reduce the rate of ICU admission, intubation, and ECMO induction. It may be necessary to be careful that order of administering antiviral drugs and steroids.

The appropriate timing of administering steroids has been unclear because this study was retrospective, not controlled. It is possible that patients exacerbated by administered steroids were extracted in this study, because all patients in the steroids-first group were transferred. Among transferred patients, twenty-three patients were in antiviral-drugs-first group. Comparing antiviral-drugs-first group (n = 16) and steroids-first group (n = 23) among transferred

patients, the formers tended higher rate of intubation, ICU admission and ECMO induction were significantly higher in the steroids-first group than antiviral-drugs-first group (S4 Table and S3 Fig). The analysis could not eliminate the possibility that the severity of patients was biased toward the steroids-first group because it was not a randomized controlled trial. Most of the transferred patients had severe respiratory disease. Among them, patients in the steroids-first group tended to have a poor prognosis.

A meta-analysis of randomized control trials [28], showed that administering steroids 1) to patients who required mechanical ventilation reduced mortality, 2) to patients who did not require intubation did not have a significant effect on mortality, and 3) for patients who did not require oxygen increased mortality. Furthermore, the patients in whom steroids were used had a significantly lower risk of needing mechanical ventilation. Thus, the effectiveness of steroid use for patients who did not require intubation may be unclear, but steroid use for patients not requiring oxygen therapy may be harmful. Authors of the meta-analysis stated the initiation of oxygen therapy and drugs at significantly different between studies as a limitation. According to RECOVERY trial [3], dexamethasone was administered at a dose of 6 mg once a day for up to 10 days. In some cases, methylprednisolone, prednisolone, or hydrocortisone were administered. In addition to the timing of steroid administration, the appropriate duration, and doses of steroids are also unclear. Pinzon et al. [29] reported that high-dose methylprednisolone for three days followed by oral prednisone for 14 days decreased recovery time and need for intensive care compared with 6 mg dexamethasone for 7 to 10 days. Mishra et al. [27] pointed out that long-term steroid use for COVID-19 may cause adverse drug reactions such as thrombosis. Further case accumulation will be required to clarify the appropriate timing, duration, and doses of steroids and combination of drugs, including steroids and their order of use.

## Study limitations

The present study had four main limitations. First, it had a retrospective single-center cohort design and low statistical power Second, the criteria for the introduction of specific drugs for COVID-19 was not standardized, so there may be confounding by indication. This was unavoidable because the treatment protocols had not yet been established and were in a state of flux. Third, most of the patients in this study were Japanese, and the virus variant were not studied. It is possible that the effect of steroids may vary by race or viral strain. Fourth, some patients were administered steroids other than dexamethasone, and the types and/or dosages of steroids may affect the outcome.

## Conclusions

In patients with early COVID-19, administering steroids prior to initiating antiviral drugs in the first few days after symptom onset, may aggravate respiratory disease severity. In patients with early COVID-19, it may be safer to administer antiviral drugs in the early phase of the disease and delay the administration of steroids.

## Supporting information

**S1 Fig. Rate of ICU admission, intubation, ECMO induction, and mortality in patients who administered steroid before 10 days of onset and those who administered steroid after 11 days of onset in the antiviral-drugs-first group.** There are the tendencies in which the rate of ICU admission, intubation was poor in patients who administered steroid before 10 days. ECMO, Extracorporeal membrane oxygenation. ICU, Intensive care unit.
(TIF)

**S2 Fig. Rate of ICU admission, intubation, ECMO induction, and mortality in the antiviral-drugs-first group and the steroids-first group in cases where steroids were administered before 10 days of onset.** The rate of ICU admission, intubation, ECMO induction were poorer in patients of the steroids-first group than in those the antiviral-drugs-first group (p = 0.007, p = 0.007 and p = 0.037, respectively, log-rank test). ECMO, Extracorporeal membrane oxygenation. ICU, Intensive care unit.
(TIF)

**S3 Fig. Rate of ICU admission, intubation, ECMO induction, and mortality in the antiviral-drugs-first group and the steroids-first group in transferred cases.** There are the tendencies in which the rate of ICU admission, intubation, ECMO induction were poor in patients of steroids-first group. ECMO, Extracorporeal membrane oxygenation. ICU, Intensive care unit.
(TIF)

**S1 Table. Comparison between the group in whom steroids administered before 10 days and after 11 days in the antiviral-drugs-first group.**
(DOCX)

**S2 Table. Comparison between the antiviral-drugs-first group and the steroids-first group in cases where steroids were administered before 10 days of onset.**
(DOCX)

**S3 Table. The time difference between antiviral drugs administration to dexamethasone administration.**
(DOCX)

**S4 Table. Comparison between the antiviral-drugs-first group and the steroids-first group in transferred cases.**
(DOCX)

**S5 Table. The source of the data from the medical records.**
(XLSX)

## Acknowledgments

We would like to thank Editage (www.editage.com) for English language editing.

We would like to thank two statisticians (Yuki Shiko, MS and Yoshihito Ozawa, BS. Biostatistics Section, Clinical Research Center, Chiba University Hospital) for help with the statistical analysis of the study data.

## Author Contributions

**Conceptualization:** Toshibumi Taniguchi.

**Data curation:** Yu Shionoya, Hajime Kasai, Noriko Sakuma, Shun Imai, Kohei Shikano, Shin Takayanagi, Misuzu Yahaba.

**Formal analysis:** Yu Shionoya, Hajime Kasai.

**Investigation:** Yu Shionoya, Hajime Kasai, Noriko Sakuma, Shun Imai, Kohei Shikano, Shin Takayanagi, Misuzu Yahaba, Taka-aki Nakada, Hidetoshi Igari, Seiichiro Sakao.

**Methodology:** Toshibumi Taniguchi.

**Project administration:** Toshibumi Taniguchi, Seiichiro Sakao.

**Supervision:** Toshibumi Taniguchi, Taka-aki Nakada, Hidetoshi Igari, Seiichiro Sakao, Takuji Suzuki.

**Writing – original draft:** Yu Shionoya, Hajime Kasai.

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
