## [Decision Letter · Decision Letter 0]

12 Jul 2021

PONE-D-21-18432

Possibility of deterioration of respiratory status when steroids precede antiviral drugs in patients with COVID-19 pneumonia

PLOS ONE

Dear Dr. Kasai,

Thank you for submitting your manuscript to PLOS ONE. After careful consideration, we feel that it has merit but does not fully meet PLOS ONE’s publication criteria as it currently stands. Therefore, we invite you to submit a revised version of the manuscript that addresses the points raised during the review process.

We look forward to receiving your revised manuscript.

Kind regards,

Aleksandar R. Zivkovic

Academic Editor

PLOS ONE

2. In the ethics statement in the manuscript and in the online submission form, please provide the source of the data analyzed in this work (e.g. medical records/database and hospital, institution or medical center name).

 [he funders had no role in study design, data collection and analysis, decision to publish, or preparation of the manuscript.].

[I have read the journal's policy and the authors of this manuscript have the following competing interests:

T.T. has received honorarium for lecture from Gilead Sciences Inc., Janssen Pharmaceuticals Inc., ViiV Healthcare Limited, and MSD Limited, respectively.

T.T. is the member of advisory board of Janssen Pharmaceuticals Inc., ViiV Healthcare Limited, and MSD Limited, respectively.].

Reviewers' comments:

Reviewer #1: This was a retrospective study; hence, other treatment modalities could have been different in the two groups, e.g. anticoagulants. Did all or some of the patients receive anticoagulants / Was anticoagulant a part of treatment protocol? The authors need to comment on this. Line no. 171, "..incidence of intubation and ECMO.." should be changed to ".. the duration of intubation and ECMO.." as per the data presented in table 2.

The following issues may be commented on / added in the discussion section. Along with the duration of symptom onset, clinical evaluation/hypoxia is also an essential factor to decide regarding initiation of steroids in Covid-19 infection. However, today's dilemma is that robust evidence regarding the absolute beneficial role of antivirals in covid-19 is lacking. No specific drugs have been approved for COVID-19 disease to date. (1) Other than the timing of corticosteroid, duration of corticosteroid therapy in covid-19 infection is also an important factor, and as evidenced from the RECOVERY trial, the patients were administered steroid therapy up to 10 days. (2) Also, corticosteroids may potentially have a procoagulant effect in covid-19; hence the addition of anticoagulant therapy may also be considered in the treatment protocol of covid-19. This could have been a potential factor, wherein steroid therapy aggravated the treatment outcomes in the absence of an antiviral, which could potentially have controlled the viremia associated procoagulant effect of the covid-19 infection at an earlier stage of the disease. (3)

The study presents a very logical finding that steroid therapy without the cover of antiviral therapy may be detrimental.

References:

1. Frediansyah A, Tiwari R, Sharun K, Dhama K, Harapan H. Antivirals for COVID-19: A critical review [Internet]. Vol. 9, Clinical Epidemiology and Global Health. Elsevier B.V.; 2021 [cited 2021 Jun 24]. p. 90–8. Available from: https://pubmed.ncbi.nlm.nih.gov/33521390/

2. Dexamethasone in Hospitalized Patients with Covid-19 — Preliminary Report. N Engl J Med [Internet]. 2020 Jul 17 [cited 2020 Oct 30]; Available from: https://www.nejm.org/doi/full/10.1056/NEJMoa2021436

3. Mishra GP, Mulani J. Corticosteroids for COVID-19: the search for an optimum duration of therapy. Lancet Respir Med [Internet]. 2021 Jan 1 [cited 2021 Jan 1];9(1):e8. Available from: https://doi.org/10.1016/S2213-2600(20)30530-0

Reviewer #2: The authors report on an interesting and relevant topic with respect tot the ongoing pandemic. However, a few fundamental flaws are present in the manuscript

1. The manuscript report on the possibility of harm when steroid use precede antiviral drugs in patients with COVID 19 disease, without elaborating the mechanism supported by data .

2.The title does not provide the study design .

3.Introduction:Manuscript mention that in RECOVERY trial , use of Dexamethasone administration among patient who did not receive any respiratory support was unclear.Its not understandable that how a retrospective design would answer this question and trump a RCT.

4.Manuscript mentions that early administration of steroids may provoke viral replication and cytokine storm of greater severity resulting in more sever respiratory failure. The RECOVERY trial, in there pre specified subgroup analysis, did not find the above observation-much more larger sample size and better design (point estimate 1.01, CI 0.87,1.17).

5. Authors cite a meta analysis and conclude that combination therapy was more effective than monotherpy is incorrect, The meta analysis looked at role of steroid alone and found its association with reduced mortality.(ref 7).

Methods :

1.The exclusion of patients who were symptomatic and initially treated with steroids is unclear. Why were they excluded?How there inclusion could have changed the study estimates ?.

2.Authors have a policy to start antiviral drugs first based on severity and risk factor . That means more sicker patients or patients with risk factors must have received antiviral and steroids within the short span of time with further deterioration , its unclear how prognostic balance was achieved in both the group for comparison without randomization .

3.The division of patients into two groups - S-A and A-S, is arbitrary as manuscript does not tell us what was the time period difference between the two drugs . Was it 24 hour /48 hours/72 hours ?

4. Authors use Kaplan Meier survival method for analysis .The basis for its use is unclear as the outcome variables-admission rate , intubation rate , echo introduction rate are short duration endpoints and knowing time to event is not helpful in understanding the incidence.

5. Result

The S-A group ended up with too low numbers (n=16) to draw any meaningful conclusion.

There is a prognostic imbalance between the two groups - The lymphocyte count was too low for the S-A group , along with other markers. This creates unequal groups for comparison and unstable estimates.

6. Discussion needs more elaborate evaluation of the author’s study findings and clarity which is missing .

7.The study’s primary finding is-In patients with early COVID-19, administering steroids prior to initiating antiviral drugs in the first few days after symptom onset, may aggravate respiratory  disease severity.However, due to major limitations in study design and low numbers , a meaningful conclusion appears difficult to draw.

Reviewer #3: Nice manuscript on a hot topic

MAJOR comments to improve the manuscript

1

Remove a S-A group and A-S group in the abstract

Call them in a simple, explicative way

Throughout all the manuscript, tables and figures, please

2

Early corticosteroids (patients not receiving oxygen) increase mortality in covid19 patients according to a meta-analysis of RCTs

https://pubmed.ncbi.nlm.nih.gov/33298370/

please extensively discuss this manuscript in the discussion

3

p9 line 171

ECMO is increased and few lines later it is not increased anymore. Please fix

4

Mortality is increased when using steroids first 18% vs 13%

The difference is not statistically significant, but there is a difference

There is no need to underline there is no difference

Please remove from the abstract

, but there was no significant difference in mortality

6. PLOS authors have the option to publish the peer review history of their article (what does this mean?). If published, this will include your full peer review and any attached files.

Reviewer #1: **Yes: **Gyanshankar P Mishra

Reviewer #2: No

Reviewer #3: No

---

## [Author Response · Author response to Decision Letter 0]

18 Aug 2021

Dear Dr. Zivkovic

Ref. No.: PONE-D-21-18432

Thank you for your email message dated July 12, 2021, regarding our manuscript, “Possibility of deterioration of respiratory status when steroids precede antiviral drugs in patients with COVID-19 pneumonia.”

We have revised our manuscript accordingly and have provided a point-by-point response to the reviewers’ comments. This is attached herewith. The changes to the manuscript are shown in red font.

We have corrected the ethics approval number from 11111 to 3929.

We added following sentence in Study design and materials of Material and methods: The source of the data analyzed in this study was the medical records, Chiba University Hospital.

We believe that our revised manuscript has suitably incorporated the reviewers’ suggestions and is significantly improved over our initial submission. We trust that it is now suitable for publication in PLOS ONE.

Thank you for considering our paper for publication.

Sincerely,

Response to Reviewer #1’s comments

Response: 

We wish to express our strong appreciation for your insightful comments on our manuscript. The comments have helped us to significantly improve the manuscript. We have marked the relevant changes in red that so you can easily find them.

Comment #1: This was a retrospective study; hence, other treatment modalities could have been different in the two groups, e.g. anticoagulants. Did all or some of the patients receive anticoagulants / Was anticoagulant a part of treatment protocol? 

Response: 

Thank you for this comment.

According to our institute’s protocol, when D-dimer is > 3.0 mg/dL, we administered direct oral anticoagulants for non-ICU patients. Patients with a hemorrhagic diathesis or elderly patients were not administered anticoagulants (in two cases). Two patients who originally taken aspirin, we continued aspirin in non-ICU patients. There were no patients who had originally taken direct oral anticoagulants. For ICU patients we administered heparin routinely. 

There was no significant difference in the anticoagulant or aspirin use between the steroids-first group (S-A group) and the antiviral-drugs-first group (A-S group) (n = 13, 81.3% vs. n = 32, 62.8% p=0.169).

Based on a comment by Reviewer 3, we revised the name “S-A group” to “steroids-first group” and “A-S group” to “antiviral-drugs-first group.”

Additionally, we administered antibiotics to some patients who were suspected to have complications of bacterial pneumonia. These drugs did not have any influence on COVID-19; hence we do not think that they affected the results of our study.

Accordingly, we added the following wording to the methods section and Table 2.

(Page 7, lines 121-127)

Added: “According to our institute’s protocol, when the D-dimer level was > 3.0 mg/dL, we administered direct oral anticoagulants in non-ICU patients, but withheld anticoagulants in elderly patients and patients with a hemorrhagic diathesis (two patients). We continued aspirin in in non-ICU patients who had originally taken aspirin (two patients). There were no patients who had originally taken direct oral anticoagulants. We administered heparin routinely to ICU patients.”

Comment #2: The authors need to comment on this. Line no. 171, “...incidence of intubation and ECMO." should be changed to “... the duration of intubation and ECMO." as per the data presented in table 2.

Response: 

According to the comment, we have reworded the related sentence as follows:

(Page 10, line 182)

Original: “In contrast, there was no significant difference in mortality, the duration of ICU admission, or the incidence of intubation and ECMO between the two groups.”

Revised: “In contrast, there was no significant difference in the duration of ICU admission, or the duration of intubation and ECMO between the two groups.”

Comment #3: The following issues may be commented on / added in the discussion section. Along with the duration of symptom onset, clinical evaluation/hypoxia is also an essential factor to decide regarding initiation of steroids in Covid-19 infection. However, today's dilemma is that robust evidence regarding the absolute beneficial role of antivirals in covid-19 is lacking. No specific drugs have been approved for COVID-19 disease to date. (1) Other than the timing of corticosteroid, duration of corticosteroid therapy in covid-19 infection is also an important factor, and as evidenced from the RECOVERY trial, the patients were administered steroid therapy up to 10 days. (2) Also, corticosteroids may potentially have a procoagulant effect in covid-19; hence the addition of anticoagulant therapy may also be considered in the treatment protocol of covid-19. This could have been a potential factor, wherein steroid therapy aggravated the treatment outcomes in the absence of an antiviral, which could potentially have controlled the viremia associated procoagulant effect of the covid-19 infection at an earlier stage of the disease. (3)

The study presents a very logical finding that steroid therapy without the cover of antiviral therapy may be detrimental.

References:

1. Frediansyah A, Tiwari R, Sharun K, Dhama K, Harapan H. Antivirals for COVID-19: A critical review [Internet]. Vol. 9, Clinical Epidemiology and Global Health. Elsevier B.V.; 2021 [cited 2021 Jun 24]. p. 90–8. Available from: https://pubmed.ncbi.nlm.nih.gov/33521390/

2. Dexamethasone in Hospitalized Patients with Covid-19 — Preliminary Report. N Engl J Med [Internet]. 2020 Jul 17 [cited 2020 Oct 30]; Available from: https://www.nejm.org/doi/full/10.1056/NEJMoa2021436

3. Mishra GP, Mulani J. Corticosteroids for COVID-19: the search for an optimum duration of therapy. Lancet Respir Med [Internet]. 2021 Jan 1 [cited 2021 Jan 1];9(1): e8. Available from: https://doi.org/10.1016/S2213-2600(20)30530-0

Response: 

Thank you for this insightful feedback and recommendation of useful reports.

As you pointed, we have also been aware of the lack of evidence for antiviral drugs and the potential impact of the double-edged sword of steroids in COVID-19 throughout our clinical practice. Your recommended issues were useful to further highlight the results of our study.

Accordingly, we added following wording:

(Page 12, lines 213-224)

Added: “In this study, remdesivir (n = 45, 67.1%) and favipiravir (n = 38, 56.7%) were used and other antiviral drugs, such as lopinavir/ritonavir, were not. Several antiviral drugs have been assessed for COVID-19 [9], but remdesivir is the only Food and Drug Administration-approved drug for the treatment of COVID-19, based on the result of ACTT-1 trial [10]. Routine use of remdesivir is not recommended according to the results of some studies and the National Institutes of Health guidelines [11-13]. In cases where patients are at a particularly high risk of clinical deterioration or require minimal supplemental oxygen, remdesivir administration should be considered [11]. Our institute’s protocol is to initiate antiviral drugs according to the severity and risk factors. In this study, the differences in the antiviral drugs used may have affected the prognosis; however, there is a lack of strong evidence regarding the benefits of antiviral drugs in COVID-19.”

(Page 13, line 244 to page 14, line 256)

Original: “In some cases, administered steroid, including dexamethasone administration, early in the viral response phase could exacerbate the disease and increase the need for critical care.”

Revised: “A higher viral load leads to apoptosis of pneumocytes and endothelial cells, which in turn activates platelets and induces coagulation factors [24]. COVID-19 has a high rate of thromboembolic complications [25], and thromboembolic complications are associated with poor prognosis [26]. Furthermore, Mishra et al. [27] pointed out that steroid use itself may be associated with thromboembolic complications because steroids tend to increase clotting factors and fibrinogen concentrations. This pathophysiology in COVID-19 is poorly understood. Early administration of steroids may be associated with a higher viral load, leading to an exacerbation of the respiratory condition, and a possible increase in the risk of thromboembolic complications, and so steroid use may increase the need for critical care in some patients with COVID-19.”

(Page 15, line 290 to page 16, line 300)

Original: “Further case accumulation will be required to clarify the appropriate timing and order of administering steroids.”

Revised: “According to RECOVERY trial [3], dexamethasone was administered at a dose of 6 mg once a day for up to 10 days. In some cases, methylprednisolone, prednisolone, or hydrocortisone were administered. In addition to the timing of steroid administration, the appropriate duration and doses of steroids are also unclear. Pinzon et al. [29] reported that high-dose methylprednisolone for three days followed by oral prednisone for 14 days decreased recovery time and need for intensive care compared with 6 mg dexamethasone for 7 to 10 days. Mishra et al. [27] pointed out that long-term steroid use for COVID-19 may cause adverse drug reactions such as thrombosis. Further case accumulation will be required to clarify the appropriate timing, duration, and doses of steroids and combination of drugs, including steroids and their order of use.”

 

Response to Reviewer #2’s comments

Comment: The authors report on an interesting and relevant topic with respect to the ongoing pandemic. However, a few fundamental flaws are present in the manuscript

Response: 

We appreciate that you consider our report to be interesting and we wish to express our strong appreciation for this comment. 

We have marked the respective changes with underlined text to make them easily identifiable.

Comment #1. The manuscript report on the possibility of harm when steroid use precede antiviral drugs in patients with COVID 19 disease, without elaborating the mechanism supported by data.

Response: 

Thank you for raising this concern. 

As you pointed out, because our study is a small number retrospective study, we regret that our discussion of the mechanism of steroids use for COVID-19 was insufficient due to a lack of data.

Therefore, we have added the results of the analysis, and new references to the discussion based on your comments and the other reviewers' comments.

Additionally, we provide more details in our responses to each comment below.

Comment #2. The title does not provide the study design.

Response: 

In accordance with the reviewer’s comment, we have revised the title as follows:

(p1. Title)

Original: “Possibility of deterioration of respiratory status when steroids precede antiviral drugs in patients with COVID-19 pneumonia”

Revised: “Possibility of deterioration of respiratory status when steroids precede antiviral drugs in patients with COVID-19 pneumonia: a retrospective study” 

Comment #3. Introduction: Manuscript mention that in RECOVERY trial, use of Dexamethasone administration among patient who did not receive any respiratory support was unclear. It’s not understandable that how a retrospective design would answer this question and trump a RCT.

Response: 

Thank you for this important comment. 

Our results did not validate the effect of dexamethasone in patients without respiratory support, but rather the effects of early administration of steroids and sequencing with antiviral drugs in COVID-19. In the RECOVERY trial, any apparent significance of dexamethasone at less than seven days of onset was not mentioned. Furthermore, in Japan, the results of the RECOVERY trial led to the insurance approval of dexamethasone, although there are no definite rules regarding the timing of administration of dexamethasone. As a result, dexamethasone has tended to be liberally administered at the discretion of clinicians even in early onset or mild cases of COVID-19. 

Accordingly, we modified the relevant sentences as follows:

(Page 4, lines 63-71)

Original: “However, there was no clear effect of dexamethasone administration among patients who did not receive any respiratory support. Thus, the timing of drug administration, including antiviral drugs, is unclear and the most effective therapeutic approach has not been determined.”

Revised: “However, the results of the RECOVERY trial did not mention any apparent significance of dexamethasone given within 7 days of onset in COVID-19. Thus, the timing of drug administration, including antiviral drugs, is unclear and the most effective therapeutic approach has not been determined. Additionally, in Japan, the results of the RECOVERY trial led to the insurance approval of dexamethasone, although the Japanese guidelines for COVID-19 treatment do not provide definite rules regarding the timing of administration of dexamethasone. As a result, dexamethasone tends to be liberally administered at the discretion of clinicians, even in early onset or mild cases of COVID-19.”

Comment #4. Manuscript mentions that early administration of steroids may provoke viral replication and cytokine storm of greater severity resulting in more severe respiratory failure. The RECOVERY trial, in there pre specified subgroup analysis, did not find the above observation-much more larger sample size and better design (point estimate 1.01, CI 0.87,1.17).

Response: 

Thank you for this comment.

As your comment, we thought the following sentence was overstatement, then we removed it, “Thus, early administration of steroids may provoke more viral replication and cytokine storms of greater severity, resulting in more severe respiratory failure.”

Accordingly, we removed following sentences.

(Page 5, lines 81-83)

Original: “In the RECOVERY Trial, dexamethasone was found to be beneficial to patients who were treated more than 7 days after symptom onset [3]. Thus, early administration of steroids may provoke more viral replication and cytokine storms of greater severity, resulting in more severe respiratory failure.”

Revised: “In the RECOVERY Trial, dexamethasone was found to be beneficial to patients who were treated more than 7 days after symptom onset [3].”

The RECOVERY trial did not show that the positive efficacy of administering steroids less than seven days after symptom onset.

As stated previously, COVID-19 has a viral response phase and a host inflammatory response phase. Some studies of SARS or Middle East respiratory syndrome reported the use of steroids may delay virus clearance and be associated higher plasma viral load. This was not about COVID-19 and the report showed that steroids use for COVID-19 did not delay virus clearance. However, in COVID-19 patients, administered steroids early in the disease may have delayed virus clearance and steroid use may be associated with a higher plasma viral load. Some reports have shown that higher viral load might be related to severity in COVID-19. This pathophysiology in COVID-19 is poorly understood. The early administration of steroids may be associated with a higher viral load, then the respiratory condition may deteriorate and the need for critical care may increase in some COVID-19 patients.

Accordingly, we removed the following sentence in Introduction and added the following sentence in Discussion.

(Page 13, line 239 to page 14, line 256)

Original: “While Liu et al., reported that administration of steroid does not affect clearance of SARS-CoV-2 [9], Russell et al., reported that the use of anti-inflammatory therapy early in the course of the disease may not have a therapeutic effect and can induce increased viral replication in patients with influenza and SARS [10]. In some cases, administered steroid, including dexamethasone administration, early in the viral response phase could exacerbate the disease and increase the need for critical care.”

Revised: “Some studies of SARS or Middle East respiratory syndrome found that steroid use may delay virus clearance and be associated higher viral concentrations [14, 15]. Furthermore, Lee et al. [15] did not recommend early steroid use without effective antiviral drugs in SARS. However, the study by Lee et al. was not about COVID-19 and some studies reported that the administration of steroids to patients with COVID-19 may not affect clearance of SARS-CoV-2 [16-21]. Evidence of the effect of steroid use on clearance of SARS-CoV-2 is currently inconclusive. Some reports showed the viral load may correlate with the severity of COVID-19 [22, 23]. A higher viral load leads to apoptosis of pneumocytes and endothelial cells, which in turn activates platelets and induces coagulation factors [24]. COVID-19 has a high rate of thromboembolic complications [25], and thromboembolic complications are associated with poor prognosis [26]. Furthermore, Mishra et al. [27] pointed out that steroid use itself may be associated with thromboembolic complications because steroids tend to increase clotting factors and fibrinogen concentrations. This pathophysiology in COVID-19 is poorly understood. Early administration of steroids may be associated with a higher viral load, leading to an exacerbation of the respiratory condition, and a possible increase in the risk of thromboembolic complications, and so steroid use may increase the need for critical care in some patients with COVID-19.”

Comment #5. Authors cite a meta analysis and conclude that combination therapy was more effective than monotherpy is incorrect, The meta analysis looked at role of steroid alone and found its association with reduced mortality.(ref 7).

Response: 

Thank you for this comment and we apologize for our mistake.

Ref 7 was wrong, and Ref 6 was correct, so we revised the sentences and the reference number in the manuscript. 

Original: “A World Health Organization working group reported that a meta-analysis showed that combination therapy was more effective than monotherapy for treating COVID-19 [7].”

Revised: “Cantini et al. [6] reported that a meta-analysis showed that combination therapy was more effective than monotherapy for treating COVID-19.”

Methods:

Comment #1. The exclusion of patients who were symptomatic and initially treated with steroids is unclear. Why were they excluded? How their inclusion could have changed the study estimates?

Response:

Thank you for this comment.

We apologize for our confusing description. In this study, patients who received only symptomatic treatment or those who were originally treated with steroids or immunosuppressive drugs because of their complications, such as rheumatoid arthritis were excluded.

This retrospective study aimed to determine the effectiveness of specific drugs for COVID-19 such as antiviral drugs, steroids, and anti-inflammatories, hence we excluded patients who originally treated with steroids or immunosuppressive drugs because of complications. 

Accordingly, we revised the following sentence.

(Page 6, lines 103-104)

Original: “Patients who received only symptomatic treatment and those who were initially treated with steroids or immunosuppressive drugs were excluded.”

Revised: “Patients who received only symptomatic treatment or those who were originally treated with steroids or immunosuppressive drugs because of their complications were excluded.”

(Page 8, lines 149-150)

Original: “Of these patients, 129 were excluded because they received only symptomatic treatment (n = 117) or were using steroids or immunosuppressive drugs (n = 12).”

Revised: “Of these patients, 129 were excluded because they received only symptomatic treatment (n = 117) or were originally using steroids or immunosuppressive drugs because of complications (n = 12).”

Comment #2. Authors have a policy to start antiviral drugs first based on severity and risk factor. That means more sicker patients or patients with risk factors must have received antiviral and steroids within the short span of time with further deterioration, it’s unclear how prognostic balance was achieved in both the group for comparison without randomization.

Comment #3. The division of patients into two groups - S-A and A-S, is arbitrary as manuscript does not tell us what was the time period difference between the two drugs. Was it 24 hour /48 hours/72 hours?

Response: 

Thank you for these comments.

We have been aware of that this study had the limitation of confirming our conclusion from the results because this study was not a randomized controlled trial but a retrospective study.

While this study was not a randomized control trial and small population, baseline characteristics that are shown in Table 1 were not different between two groups other than age. The antiviral-drugs-first group (A-S group*) being older than the steroid-first group (S-A group) is a limitation of this study. However, the antiviral-first group had a better prognosis, and the other risk factors were not different between the two groups. Therefore, we think that the risk factors did not contribute to the difference in prognosis between groups in this study.

*Based on Reviewer #3’s comments, we revised S-A group to steroids-first group and A-S group to antiviral-drugs-first group.

Accordingly, we added the following wording:

(Page 12, lines 205-208)

Added: “Older age is known to be a risk factor exacerbating COVID-19 [8]. The antiviral-drugs-first group were significantly older than the steroids-first group. Despite the antiviral drugs-first group being older than the steroid-first group, the antiviral drugs-first group had a better prognosis.

According to our institute’s protocol, if respiratory failure worsened steroids were added to patients who had been administered antiviral drugs already. If a patient needs fraction of inspiratory oxygen > 0.4 at the time of admission, antiviral drugs and dexamethasone were administered almost the same time, but antiviral drugs were administered 3 to 6 hours ahead.

Additionally, the average time of administered antiviral drugs to steroids were 2.7 ± 1.9 days.

As you pointed out, we analyzed ICU admission rate, incubation rate, ECMO induction rate between two groups depending on the time difference between antiviral drugs administration to dexamethasone administration within 24 hours, 48 hours, and 72 hours.

The results show in new S3 table.

New S3 table. The time difference between antiviral drugs administration to dexamethasone administration.

Parameter Steroids-first group

(N=16) Antiviral-drugs-first group (N=51)

 within 24 hours

(n=33) within 48 hours

(n=40) within 72 hours

(n=44)

ICU admission, n (%) 13 (81.3) 13 (39.4) † 16 (46.9) † 16 (36.4) †

Intubation, n (%) 12 (75.0) 11 (33.3) † 14 (35.0) † 14 (31.8) †

ECMO, n (%) 5 (31.3) 3 (9.1) * 3 (7.5) * 3 (6.8) *

ECMO, Extracorporeal membrane oxygenation. ICU, Intensive care unit.

versus Steroids-first group, * P<0.05, † P<0.01

The patients in the antiviral-drugs-first group had a lower rate of ICU admission, incubation. and ECMO induction than patients in steroids-first group in all timings of administration of steroids. The patients with shorter time differences between antiviral drug and steroid initiation may have been sicker patients. Comparison between them and patients in steroid-first group, the patients in the antiviral-drugs-first group had a better prognosis.

This study was not randomized control study; thus, it is possible that sicker patients tend to be treated with steroids first. However, from our results, it appears that administering antiviral drugs first might be better than administering steroids first.

Accordingly, we revised and added wording to the manuscript. Furthermore, we added new S3 table.

(Page 6, line 117 to page 7, line 1)

Original: “Then, if respiratory failure worsened, steroids (mostly dexamethasone) were administered.”

Revised: “Then, if respiratory failure worsened, steroids (mostly dexamethasone) were administered, and if patients needed a fraction of inspiratory oxygen > 0.4 at the time of admission, antiviral drugs and dexamethasone were initiated at almost the same time, but antiviral drugs were initiated 3 to 6 hours ahead of dexamethasone.”

(Page 14, lines 265-267)

Added: “Furthermore, patients who received short-term steroids also had a significantly lower rate of ICU admission, intubation, and ECMO induction than patients in the steroid-first group (S3 Table).”

Comment #4. Authors use Kaplan Meier survival method for analysis. The basis for its use is unclear as the outcome variables-admission rate, intubation rate, echo introduction rate are short duration endpoints and knowing time to event is not helpful in understanding the incidence.

Response: 

Thank you for this comment.

COVID-19 generally progress rapidly. ICU admission, intubation, and ECMO induction caused by COVID-19 occurs in a short period. In fact, the primary outcome of the RECOVERY trial was 28-day mortality and, 64.8% of the patients were discharged within 28 days in the RECOVERY trial. In our study, median time from symptom onset of ICU admission was 9 days (n = 30, range 6 to 19); to intubation, 9 days (n = 27, range 6 to 19); to ECMO induction, 9 days (n = 9, range 8 to 21). Therefore, we decided that the Kaplan-Meier survival method applied to these events that occurred within 30 days and set discharge or death as censoring events using this method. 

We have subsequently consulted statisticians in our institute, who are mentioned in the acknowledgements (Yuki Shiko, MS and Yoshihito Ozawa, MS. Biostatistics Section, Clinical Research Center, Chiba University Hospital), regarding this point. They were of the same opinion.

From the above, we think that the Kaplan-Meier survival method for analysis may be useful in our study.

Comment #5a. Result: The S-A group ended up with too low numbers (n=16) to draw any meaningful conclusion.

Response:

Thank you for this comment.

We mentioned that small population and the retrospective study design as limitations in the study limitations subsection.

While it may be difficult to confirm our conclusion from this small population and retrospective study, the results of our study provide some insight on the appropriate use of steroids in COVID-19.

Comment #5b. There is a prognostic imbalance between the two groups - The lymphocyte count was too low for the S-A group, along with other markers. This creates unequal groups for comparison and unstable estimates.

Response: 

Thank you for this comment.

Patients in the steroid-first group were administered steroids before being admitted to our institute, therefore the lymphocyte count was lower than that of antiviral-drugs-first group. We analyzed the blood test results undergone at our hospital admission, because of the blood test results taken in former hospital were often incomplete.

Accordingly, we added the following text:

(Page 12, lines 208-212)

Added: “The steroid-first group patients were administered steroids before admission to our hospital, therefore their lymphocyte count was significantly lower than that of antiviral drugs-first group. The blood tests performed on admission to our hospital were analyzed, because of the blood test results performed in the original hospital were frequently incomplete.”

Comment #6. Discussion needs more elaborate evaluation of the author’s study findings and clarity which is missing.

Response: 

Thank you for this comment.

We revised the Discussion based on your comment and another reviewer’s comment.

Comment #7. The study’s primary finding is-In patients with early COVID-19, administering steroids prior to initiating antiviral drugs in the first few days after symptom onset, may aggravate respiratory disease severity. However, due to major limitations in study design and low numbers, a meaningful conclusion appears difficult to draw.

Response: 

Thank you for this comment.

The small sample size is a limitation of this study. While we do not think can confirm our conclusion from this small population and retrospective study, the results provide some useful insight.

 

Response to Reviewer #3’s comments

COMMENT: Nice manuscript on a hot topic

Response: 

We appreciate that you consider our report to be interesting and we wish to express our strong appreciation of this comment. We have marked the changes to the manuscript in response to your comments with black highlights to make them easily identifiable.

Major Comment

Comment #1. Remove a S-A group and A-S group in the abstract

Call them in a simple, explicative way

Throughout all the manuscript, tables, and figures, please

Response: 

Thank you for this comment.

As you pointed out, the group names can be confusing. Therefore, we revised S-A group to “steroids-first group” and A-S group to “antiviral-drugs-first group”, to make it easy for the reader to understand.

Accordingly, we revised the group names throughout the text, tables, figures.

Comment #2. Early corticosteroids (patients not receiving oxygen) increase mortality in covid19 patients according to a meta-analysis of RCTs

https://pubmed.ncbi.nlm.nih.gov/33298370/

Please extensively discuss this manuscript in the discussion.

Response: 

Thank you for this insightful feedback and recommendation of this useful report.

The meta-analysis of randomized control trial, including the RECOVERY trial, showed that administering steroids 1) for patients who required mechanical ventilation reduced mortality; 2) for patients who did not required intubation did not have a significant effect on mortality; and 3) for patients not requiring oxygen increased mortality. Furthermore, steroid use was associated with a significantly lower risk of need for mechanical ventilation. Thus, the effectiveness of steroid use for patients who did not require intubation is unclear, but steroid use for patients not requiring oxygen therapy may be harmful.

The authors of the meta-analysis that you recommended, noted that the initiation of oxygen therapy and drugs were significantly different between studies and reported this as a limitation.

Further study will be required to clarify the appropriate timing of administering steroid and the combination of steroids and other drugs.

Accordingly, we revised following text:

(Page 15, line 282 to page 16, line 300)

Revised: “A meta-analysis of randomized control trials [28], showed that administering steroids 1) to patients who required mechanical ventilation reduced mortality, 2) to patients who did not require intubation did not have a significant effect on mortality, and 3) for patients who did not require oxygen increased mortality. Furthermore, the patients in whom steroids were used had a significantly lower risk of needing mechanical ventilation. Thus, the effectiveness of steroid use for patients who did not require intubation may be unclear, but steroid use for patients not requiring oxygen therapy may be harmful. Authors of the meta-analysis stated the initiation of oxygen therapy and drugs at significantly different between studies as a limitation. According to RECOVERY trial [3], dexamethasone was administered at a dose of 6 mg once a day for up to 10 days. In some cases, methylprednisolone, prednisolone, or hydrocortisone were administered. In addition to the timing of steroid administration, the appropriate duration and doses of steroids are also unclear. Pinzon et al. [29] reported that high-dose methylprednisolone for three days followed by oral prednisone for 14 days decreased recovery time and need for intensive care compared with 6 mg dexamethasone for 7 to 10 days. Mishra et al. [27] pointed out that long-term steroid use for COVID-19 may cause adverse drug reactions such as thrombosis. Further case accumulation will be required to clarify the appropriate timing, duration, and doses of steroids and combination of drugs, including steroids and their order of use.”

Comment #3. p9 line 171 ECMO is increased and few lines later it is not increased anymore. Please fix

Response: 

According to the comment, we have reworded the related sentence as follows:

(Page 10, line 182)

Original: “In contrast, there was no significant difference in mortality, the duration of ICU admission, or the incidence of intubation and ECMO between the two groups.”

Revised: “In contrast, there was no significant difference in the duration of ICU admission, or the duration of intubation and ECMO between the two groups.”

Comment #4. Mortality is increased when using steroids first 18% vs 13%

The difference is not statistically significant, but there is a difference

There is no need to underline there is no difference

Please remove from the abstract, but there was no significant difference in mortality

Response: 

We wish to thank the reviewer for this comment. 

According to the comment, we have removed the sentence.

(p2. Results in Abstract)

Original: “The rate of intubation, ICU admission and ECMO induction were significantly higher in the S-A group than in the A-S group (81.3% vs. 33.3, p<0.001, 75.0% vs. 29.4%, p=0.001, and 31.3% vs. 7.8%, p=0.017, respectively), but there was no significant difference in mortality.”

Revised: “The rate of intubation, ICU admission and ECMO induction were significantly higher in the steroids first group than in the antiviral drugs first group (81.3% vs. 33.3, p<0.001, 75.0% vs. 29.4%, p=0.001, and 31.3% vs. 7.8%, p=0.017, respectively).”

(p10. Comparison of the outcomes among two groups in Results)

Original: “In contrast, there was no significant difference in mortality, the duration of ICU admission, or the incidence of intubation and ECMO between the two groups.”

Revised: “In contrast, there was no significant difference in the duration of ICU admission, or the duration of intubation and ECMO between the two groups.”

---

## [Editor Report · Decision Letter 1]

20 Aug 2021

Possibility of deterioration of respiratory status when steroids precede antiviral drugs in patients with COVID-19 pneumonia: a retrospective study

PONE-D-21-18432R1

Dear Dr. Kasai,

We’re pleased to inform you that your manuscript has been judged scientifically suitable for publication and will be formally accepted for publication once it meets all outstanding technical requirements.

Kind regards,

Aleksandar R. Zivkovic

Academic Editor

PLOS ONE

---

## [Editor Report · Acceptance letter]

24 Aug 2021

PONE-D-21-18432R1 

Possibility of deterioration of respiratory status when steroids precede antiviral drugs in patients with COVID-19 pneumonia: a retrospective study 

Dear Dr. Kasai:

I'm pleased to inform you that your manuscript has been deemed suitable for publication in PLOS ONE. Congratulations! Your manuscript is now with our production department. 

Kind regards, 

on behalf of

Dr. Aleksandar R. Zivkovic 

Academic Editor

PLOS ONE